# Efficient Online Linear Optimization
# with Approximation Algorithms

**Dan Garber**
Technion - Israel Institute of Technology
`dangar@technion.ac.il`

## Abstract

We revisit the problem of *online linear optimization* in case the set of feasible actions is accessible through an approximated linear optimization oracle with a factor $\alpha$ multiplicative approximation guarantee. This setting is in particular interesting since it captures natural online extensions of well-studied *offline* linear optimization problems which are NP-hard, yet admit efficient approximation algorithms. The goal here is to minimize the $\alpha$-*regret* which is the natural extension of the standard *regret* in *online learning* to this setting. We present new algorithms with significantly improved oracle complexity for both the full information and bandit variants of the problem. Mainly, for both variants, we present $\alpha$-regret bounds of $O(T^{-1/3})$, were $T$ is the number of prediction rounds, using only $O(\log(T))$ calls to the approximation oracle per iteration, on average. These are the first results to obtain both average oracle complexity of $O(\log(T))$ (or even poly-logarithmic in $T$) and $\alpha$-regret bound $O(T^{-c})$ for a constant $c > 0$, for both variants.

## 1 Introduction

In this paper we revisit the problem of *Online Linear Optimization* (OLO) [14], which is a specialized case of *Online Convex Optimization* (OCO) [12] with linear loss functions, in case the feasible set of actions is accessible through an oracle for approximated linear optimization with a multiplicative approximation error guarantee. In the standard setting of OLO, a decision maker is repeatedly required to choose an action, a vector in some fixed feasible set in $\mathbb{R}^d$. After choosing his action, the decision maker incurs loss (or payoff) given by the inner product between his selected vector and a vector chosen by an adversary. This game between the decision maker and the adversary then repeats itself. In the *full information* variant of the problem, after the decision maker receives his loss (payoff) on a certain round, he gets to observe the vector chosen by the adversary. In the *bandit* version of the problem, the decision maker only observes his loss (payoff) and does not get to observe the adversary's vector. The standard goal of the decision maker in OLO is to minimize a quantity known as *regret*, which measures the difference between the average loss of the decision maker on a game of $T$ consecutive rounds (where $T$ is fixed and known in advance), and the average loss of the best feasible action in hindsight (i.e., chosen with knowledge of all actions of the adversary throughout the $T$ rounds) (in case of payoffs this difference is reversed). The main concern when designing algorithms for choosing the actions of the decision maker, is guaranteeing that the regret goes to zero as the length of the game $T$ increases, as fast as possible (i.e., the rate of the regret in terms of $T$). It should be noted that in this paper we focus on the case in which the adversary is *oblivious* (a.k.a. *non-adaptive*), which means the adversary chooses his entire sequence of actions for the $T$ rounds beforehand.

While there exist well known algorithms for choosing the decision maker's actions which guarantee optimal regret bounds in $T$, such as the celebrated *Follow the Perturbed Leader* (FPL) and *Online Gradient Descent* (OGD) algorithms [14, 17, 12], efficient implementation of these algorithms hinges

on the ability to efficiently solve certain convex optimization problems (e.g., linear minimization for FPL or Euclidean projection for OGD) over the feasible set (or the convex hull of feasible points). However, when the feasible set corresponds for instance to the set of all possible solutions to some NP-Hard optimization problem, no such efficient implementations are known (or even widely believed to exist), and thus these celebrated regret-minimizing procedures cannot be efficiently applied. Luckily, many NP-Hard linear optimization problems (i.e., the objective function to either minimize or maximize is linear) admit efficient approximation algorithms with a multiplicative approximation guarantee. Some examples include MAX-CUT (factor 0.87856 approximation due to [9]) , METRIC TSP (factor 1.5 approximation due to [6]), MINIMUM WEIGHTED VERTEX COVER (factor 2 approximation [4]), and WEIGHTED SET COVER (factor $(\log n + 1)$ approximation due to [7]). It is thus natural to ask wether an efficient factor $\alpha$ approximation algorithm for an NP-Hard *offline* linear optimization problem could be used to construct, in a generic way, an efficient algorithm for the *online* version of the problem. Note that in this case, even efficiently computing the best fixed action in hindsight is not possible, and thus, minimizing regret via an efficient algorithm does not seem likely (given an approximation algorithm we can however compute in hindsight a decision that corresponds to at most (at least) $\alpha$ times the average loss (payoff) of the best fixed decision in hindsight).

In their paper [13], Kakade, Kalai and Ligett were the first to address this question in a fully generic way. They showed that using only an $\alpha$-approximation oracle for the set of feasible actions, it is possible, at a high level, to construct an online algorithm which achieves vanishing (expected) $\alpha$-regret, which is the difference between the average loss of the decision maker and $\alpha$ times the average loss of the best fixed point in hindsight (for loss minimization problems and $\alpha \geq 1$; a corresponding definition exists for payoff maximization problems and $\alpha < 1$). Concretely, [13] showed that one can guarantee $O(T^{-1/2})$ expected $\alpha$-regret in the full-information setting, which is optimal, and $O(T^{-1/3})$ in the bandit setting under the additional assumption of the availability of a *Barycentric Spanner* (which we discuss in the sequel).

While the algorithm in [13] achieves an optimal $\alpha$-regret bound (in terms of $T$) for the full information setting, in terms of computational complexity, the algorithm requires, in worst case, to perform on each round $O(T)$ calls to the approximation oracle, which might be prohibitive and render the algorithm inefficient, since as discussed, in general, $T$ is assumed to grow to infinity and thus the dependence of the runtime on $T$ is of primary interest. Similarly, their algorithm for the bandit setting requires $O(T^{2/3})$ calls to the approximation oracle per iteration.

The main contribution of our work is in providing new low $\alpha$-regret algorithms for the full information and bandit settings with significantly improved oracle complexities. A detailed comparison with [13] is given in Table 1. Concretely, for the full-information setting, we show it is possible to achieve $O(T^{-1/3})$ expected $\alpha$-regret using only $O(\log(T))$ calls to the approximation oracle per iteration, on average, which significantly improves over the $O(T)$ bound of [13][1]. We also show a bound of $O(T^{-1/2})$ on the expected $\alpha$-regret (which is optimal) using only $O(\sqrt{T}\log(T))$ calls to the oracle per iteration, on average, which gives nearly quadratic improvement over [13]. In the bandit setting we show it is possible to obtain a $O(T^{-1/3})$ bound on the expected $\alpha$-regret (same as in [13]) using only $O(\log(T))$ calls to the oracle per iteration, on average, under the same assumption on the availability of a *Barycentric Spanner* (BS). It is important to note that while there exist algorithms for OLO with bandit feedback which guarantee $\tilde{O}(T^{-1/2})$ expected regret [1, 11] (where the $\tilde{O}(\cdot)$ hides poly-logarithmic factors in $T$), these require on each iteration to either solve to arbitrarily small accuracy a convex optimization problem over the feasible set [1], or sample a point from the feasible set according to a specified distribution [11], both of which cannot be implemented efficiently in our setting. On the other-hand, as we formally show in the sequel, at a high level, using a BS (originally introduced in [2]) simply requires to find a single set of $d$ points from the feasible set which span the entire space $\mathbb{R}^d$ (assuming this is possible, otherwise the set could be mapped to a lower dimensional space). The process of finding these vectors can be viewed as a preprocessing step and thus can be carried out offline. Moreover, as discussed in [13], for many NP-Hard problems it is possible to compute a BS in polynomial time and thus even this preprocessing step is efficient. Importantly, [13] shows that the approximation oracle by itself is not strong enough to guarantee non-trivial $\alpha$-regret in the bandit setting, and hence this assumption on the availability of a BS seems reasonable. Since the

| | full information | | bandit information | |
|---|---|---|---|---|
| Reference | $\alpha - $ regret | oracle complexity | $\alpha - $ regret | oracle complexity |
| KKL [13] | $T^{-1/2}$ | $T$ | $T^{-1/3}$ | $T^{2/3}$ |
| This paper (Thm. 4.1, 4.2) | $T^{-1/3}$ | $\log(T)$ | $T^{-1/3}$ | $\log(T)$ |
| This paper (Thm. 4.1) | $T^{-1/2}$ | $\sqrt{T}\log(T)$ | - | - |

Table 1: comparison of expected $\alpha - $ regret bounds and average number of calls to the approximation oracle per iteration. In all bounds we give only the dependence on the length of the game $T$ and omit all other dependencies which we treat as constants. In the bandit setting we report the *expected* number of calls to the oracle per iteration.

best general regret bound known using a BS is $O(T^{-1/3})$, the $\alpha$-regret bound of our bandit algorithm is the best achievable to date via an efficient algorithm.

Technically, the main challenge in the considered setting is that as discussed, we cannot readily apply standard tools such as FPL and OGD. At a high level, in [13] it was shown that it is possible to apply the OGD method by replacing the exact projection step of OGD with an iterative algorithm which finds an *infeasible* point, but one that both satisfies the projection property required by OGD and is dominated by a convex combination of feasible points for every relevant linear loss (payoff) function. Unfortunately, in worst case, the number of queries to the approximation oracle required by this so-called projection algorithm per iteration is linear in $T$. While our online algorithms are also based on an application of OGD, our approach to computing the so-called projections is drastically different than [13], and is based on a coupling of two *cutting plane methods*, one that is based on the Ellipsoid method, and the other that resembles Gradient Descent. This approach might be of independent interest and might prove useful to similar problems.

## 1.1 Additional related work

Kalai and Vempala [14] showed that approximation algorithms which have *point-wise approximation guarantee*, such as the celebrated MAX-CUT algorithm of [9], could be used to instantiate their *Follow the Perturbed Leader* framework to achieve low $\alpha$-regret. However this construction is far from generic and requires the oracle to satisfy additional non-trivial conditions. This approach was also used in [3]. In [14] it was also shown that FPL could be instantiated with a FPTAS to achieve low $\alpha$-regret, however the approximation factor in the FPTAS needs to be set to roughly $(1 + O(T^{-1/2}))$, which may result in prohibitive running times even if a FPTAS for the underlying problem is available. Similarly, in [8] it was shown that if the approximation algorithm is based on solving a convex relaxation of the original, possibly NP-Hard, problem, this additional structure can be used with the FPL framework to achieve low $\alpha$-regret efficiently. To conclude all of the latter works consider specialized cases in which the approximation oracle satisfies additional non-trivial assumptions beyond its approximation guarantee, whereas here, similarly to [13], we will be interested in a generic as possible conversion from the offline problem to the online one, without imposing additional structure on the offline oracle.

## 2 Preliminaries

### 2.1 Online linear optimization with approximation oracles

Let $\mathcal{K}, \mathcal{F}$ be compact sets of points in $\mathbb{R}_+^d$ (non-negative orthant in $\mathbb{R}^d$) such that $\max_{\mathbf{x}\in\mathcal{K}} \|\mathbf{x}\| \leq R, \max_{\mathbf{f}\in\mathcal{F}} \|\mathbf{f}\| \leq F$, for some $R > 0, F > 0$ (throughout this work we let $\|\cdot\|$ denote the standard Euclidean norm), and for all $\mathbf{x} \in \mathcal{K}, \mathbf{f} \in \mathcal{F}$ it holds that $C \geq \mathbf{x} \cdot \mathbf{f} \geq 0$, for some $C > 0$.

We assume $\mathcal{K}$ is accessible through an approximated linear optimization oracle $\mathcal{O}_\mathcal{K} : \mathbb{R}_+^d \to \mathcal{K}$ with parameter $\alpha > 0$ such that:

$$\forall \mathbf{c} \in \mathbb{R}_+^d : \quad \mathcal{O}_\mathcal{K}(\mathbf{c}) \in \mathcal{K} \quad \text{and} \quad \begin{cases} \mathcal{O}_\mathcal{K}(\mathbf{c}) \cdot \mathbf{c} \leq \alpha \min_{\mathbf{x}\in\mathcal{K}} \mathbf{x} \cdot \mathbf{c} & \text{if } \alpha \geq 1; \\ \mathcal{O}_\mathcal{K}(\mathbf{c}) \cdot \mathbf{c} \geq \alpha \max_{\mathbf{x}\in\mathcal{K}} \mathbf{x} \cdot \mathbf{c} & \text{if } \alpha < 1. \end{cases}$$

Here $\mathcal{K}$ is the feasible set of actions for the player, and $\mathcal{F}$ is the set of all possible loss/payoff vectors[2].

Since naturally a factor $\alpha > 1$ for the approximation oracle is reasonable only for loss minimization problems, and a value $\alpha < 1$ is reasonable for payoff maximization problems, throughout this work it will be convenient to use the value of $\alpha$ to differentiate between minimization problems and maximization problems.

Given a sequence of linear loss/payoff functions $\{\mathbf{f}_1, ..., \mathbf{f}_T\} \in \mathcal{F}^T$ and a sequence of feasible points $\{\mathbf{x}_1, ...., \mathbf{x}_T\} \in \mathcal{K}^T$, we define the $\alpha - \text{regret}$ of the sequence $\{\mathbf{x}_t\}_{t \in [T]}$ with respect to the sequence $\{\mathbf{f}_t\}_{t \in [T]}$ as

$$\alpha - \text{regret}(\{(\mathbf{x}_t, \mathbf{f}_t)\}_{t \in [T]}) := \begin{cases} \frac{1}{T} \sum_{t=1}^{T} \mathbf{x}_t \cdot \mathbf{f}_t - \alpha \cdot \min_{\mathbf{x} \in \mathcal{K}} \frac{1}{T} \sum_{t=1}^{T} \mathbf{x} \cdot \mathbf{f}_t & \text{if } \alpha \geq 1; \\[2mm] \alpha \cdot \max_{\mathbf{x} \in \mathcal{K}} \frac{1}{T} \sum_{t=1}^{T} \mathbf{x} \cdot \mathbf{f}_t - \frac{1}{T} \sum_{t=1}^{T} \mathbf{x}_t \cdot \mathbf{f}_t & \text{if } \alpha < 1. \end{cases} \quad (1)$$

When the sequences $\{\mathbf{x}_t\}_{t \in [T]}, \{\mathbf{f}_t\}_{t \in [T]}$ are obvious from context we will simply write $\alpha - \text{regret}$ without stating these sequences. Also, when the sequence $\{\mathbf{x}_t\}_{t \in [T]}$ is randomized we will use $\mathbb{E}[\alpha - \text{regret}]$ to denote the expected $\alpha$-regret.

### 2.1.1 Online linear optimization with full information

In OLO with full information, we consider a repeated game of $T$ prediction rounds, for a fixed $T$, where on each round $t$, the decision maker is required to choose a feasible action $\mathbf{x}_t \in \mathcal{K}$. After committing to his choice, a linear loss function $\mathbf{f}_t \in \mathcal{F}$ is revealed, and the decision maker incurs loss of $\mathbf{x}_t \cdot \mathbf{f}_t$. In the payoff version, the decision maker incurs payoff of $\mathbf{x}_t \cdot \mathbf{f}_t$. The game then continues to the next round. The overall goal of the decision maker is to guarantee that $\alpha - \text{regret}(\{(\mathbf{x}_t, \mathbf{f}_t)\}_{t \in [T]}) = O(T^{-c})$ for some $c > 0$, at least in expectation (in fact using randomization is mandatory since $\mathcal{K}$ need not be convex). Here we assume that the adversary is *oblivious* (aka *non-adaptive*), i.e., the sequence of losses/payoffs $\mathbf{f}_1, ..., \mathbf{f}_T$ is chosen in advance (before the first round), and does not depend on the actions of the decision maker.

### 2.1.2 Bandit feedback

The bandit version of the problem is identical to the full information setting with one crucial difference: on each round $t$, after making his choice, the decision maker does not observe the vector $\mathbf{f}_t$, but only the value of his loss/payoff, given by $\mathbf{x}_t \cdot \mathbf{f}_t$.

## 2.2 Additional notation

For any two sets $\mathcal{S}, \mathcal{K} \subset \mathbb{R}^d$ and a scalar $\beta \in \mathbb{R}$ we define the sets $\mathcal{S} + \mathcal{K} := \{\mathbf{x} + \mathbf{y} \mid \mathbf{x} \in \mathcal{S}, \mathbf{y} \in \mathcal{K}\}$, $\beta\mathcal{S} := \{\beta\mathbf{x} \mid \mathbf{x} \in \mathcal{S}\}$. We also denote by $\text{CH}(\mathcal{K})$ the convex-hull of all points in a set $\mathcal{K}$. For a convex and compact set $\mathcal{S} \subset \mathbb{R}^d$ and a point $\mathbf{x} \in \mathbb{R}^d$ we define $\text{dist}(\mathbf{x}, \mathcal{S}) := \min_{\mathbf{z} \in \mathcal{S}} \|\mathbf{z} - \mathbf{x}\|$. We let $\mathcal{B}(\mathbf{c}, r)$ denote the Euclidean ball or radius $r$ centered in $\mathbf{c}$.

## 2.3 Basic algorithmic tools

We now briefly describe two very basic ideas that are essential for constructing our algorithms, namely the *extended approximation oracle* and the *online gradient descent without feasibility* method. These were already suggested in [13] to obtain their low $\alpha$-regret algorithms. We note that in the appendix we describe in more detail the approach of [13] and discuss its shortcomings in obtaining oracle-efficient algorithms.

### 2.3.1 The extended approximation oracle

As discussed, a key difficulty of our setting that prevents us from directly applying well studied algorithms for OLO, is that essentially all standard algorithms require to exactly solve (or up to arbitrarily small error) some linear/convex optimization problem over the convexification of the feasible set $\text{CH}(\mathcal{K})$. However, not only that our approximation oracle $\mathcal{O}_{\mathcal{K}}(\cdot)$ cannot perform exact minimization, even for $\alpha = 1$ it is applicable only with inputs in $\mathbb{R}_+^d$, and hence cannot optimize in all directions. A natural approach, suggested in [13], to overcome the approximation error of the oracle $\mathcal{O}_{\mathcal{K}}(\cdot)$, is to consider optimization with respect to the convex set $\text{CH}(\alpha\mathcal{K})$ (i.e. convex hull of all points in $\mathcal{K}$ scaled by a factor of $\alpha$) instead of $\text{CH}(\mathcal{K})$. Indeed, if we consider for instance the case $\alpha \geq 1$, it is straightforward to see that for any $\mathbf{c} \in \mathbb{R}_+^d$, $\mathcal{O}_{\mathcal{K}}(\mathbf{c}) \cdot \mathbf{c} \leq \alpha \min_{\mathbf{x} \in \mathcal{K}} \mathbf{x} \cdot \mathbf{c} =$

$\alpha \min_{\mathbf{x} \in \mathrm{CH}(\mathcal{K})} \mathbf{x} \cdot \mathbf{c} = \min_{\mathbf{x} \in \mathrm{CH}(\alpha\mathcal{K})} \mathbf{x} \cdot \mathbf{c}$. Thus, in a certain sense, $\mathcal{O}_{\mathcal{K}}(\cdot)$ can optimize with respect to $\mathrm{CH}(\alpha\mathcal{K})$ for all directions in $\mathbb{R}_+^d$, although the oracle returns points in the original set $\mathcal{K}$.

The following lemma shows that one can easily extend the oracle $\mathcal{O}_{\mathcal{K}}(\cdot)$ to optimize with respect to all directions in $\mathbb{R}^d$.

**Lemma 2.1** (Extended approximation oracle). *Given $\mathbf{c} \in \mathbb{R}^d$ write $\mathbf{c} = \mathbf{c}^+ + \mathbf{c}^-$ where $\mathbf{c}^+$ equals to $\mathbf{c}$ on all non-negative coordinates of $\mathbf{c}$ and zero everywhere else, and $\mathbf{c}^-$ equals $\mathbf{c}$ on all negative coordinates and zero everywhere else. The extended approximation oracle is a mapping $\hat{\mathcal{O}}_{\mathcal{K}} : \mathbb{R}^d \to (\mathcal{K} + \mathcal{B}(0, (1+\alpha)R), \mathcal{K})$ defined as:*

$$\hat{\mathcal{O}}_{\mathcal{K}}(\mathbf{c}) = (\mathbf{v}, \mathbf{s}) := \begin{cases} (\mathcal{O}_{\mathcal{K}}(\mathbf{c}^+) - \alpha R \bar{\mathbf{c}}^-, \ \mathcal{O}_{\mathcal{K}}(\mathbf{c}^+)) & \text{if } \alpha \geq 1; \\ (\mathcal{O}_{\mathcal{K}}(-\mathbf{c}^-) - R \bar{\mathbf{c}}^+, \ \mathcal{O}_{\mathcal{K}}(-\mathbf{c}^-)) & \text{if } \alpha < 1, \end{cases} \tag{2}$$

*where for any vector $\mathbf{v} \in \mathbb{R}^d$ we denote $\bar{\mathbf{v}} = \mathbf{v}/\|\mathbf{v}\|$ if $\|\mathbf{v}\| > 0$ and $\bar{\mathbf{v}} = \mathbf{0}$ otherwise, and it satisfies the following three properties:*

1. *$\mathbf{v} \cdot \mathbf{c} \leq \min_{\mathbf{x} \in \alpha\mathcal{K}} \mathbf{x} \cdot \mathbf{c}$*

2. *$\forall \mathbf{f} \in \mathcal{F}: \mathbf{s} \cdot \mathbf{f} \leq \mathbf{v} \cdot \mathbf{f}$ if $\alpha \geq 1$ and $\mathbf{s} \cdot \mathbf{f} \geq \mathbf{v} \cdot \mathbf{f}$ if $\alpha < 1$*

3. *$\|\mathbf{v}\| \leq (\alpha + 2)R$*

The proof is given in the appendix for completeness.

It is important to note that while the extended oracle provides solutions with values at least as low as any point in $\mathrm{CH}(\alpha\mathcal{K})$, still in general the output point $\mathbf{v}$ need not be in either $\mathcal{K}$ or $\mathrm{CH}(\alpha\mathcal{K})$, which means that it is not a feasible point to play in our OLO setting, nor does it allow us to optimize over $\mathrm{CH}(\alpha\mathcal{K})$. This is why we also need the oracle to output the feasible point $\mathbf{s} \in \mathcal{K}$ which dominates $\mathbf{v}$ for any possible loss/payoff vector in $\mathcal{F}$. While we will use the outputs $\mathbf{v}$ to solve a certain optimization problem involving $\mathrm{CH}(\alpha\mathcal{K})$, this dominance relation will be used to convert the solutions to these optimization problems into feasible plays for our OLO algorithms.

### 2.3.2 Online gradient descent with and without feasibility

As in [13], our online algorithms will be based on the well known *Online Gradient Descent* method (OGD) for online convex optimization, originally due to [17]. For a sequence of loss vectors $\{\mathbf{f}_1, ..., \mathbf{f}_T\} \subset \mathbb{R}^d$ OGD produces a sequence of plays $\{\mathbf{x}_1, ..., \mathbf{x}_T\} \subset \mathcal{S}$, for a convex and compact set $\mathcal{S} \subset \mathbb{R}^d$ via the following updates: $\forall t \geq 1 : \mathbf{y}_{t+1} \leftarrow \mathbf{x}_t - \eta\mathbf{f}_t, \mathbf{x}_{t+1} \leftarrow \arg\min_{\mathbf{x} \in \mathcal{S}} \|\mathbf{x} - \mathbf{y}_{t+1}\|^2$, where $\mathbf{x}_1$ is initialized to some arbitrary point in $\mathcal{S}$ and $\eta$ is some pre-determined step-size. The obvious difficulty in applying OGD to online linear optimization over $\mathcal{S} = \mathrm{CH}(\alpha\mathcal{K})$ is the step of computing $\mathbf{x}_{t+1}$ by projecting $\mathbf{y}_{t+1}$ onto the feasible set $\mathcal{S}$, since as discussed, even with the extended approximation oracle, one cannot exactly optimize over $\mathrm{CH}(\alpha\mathcal{K})$. Instead we will consider a variant of OGD which may produce infeasible points, i.e., outside of $\mathcal{S}$, but which guarantees low regret with respect to any point in $\mathcal{S}$. This algorithm, which we refer to as *online gradient descent without feasibility*, is given below (Algorithm 1).

---

**Algorithm 1** Online Gradient Descent Without Feasibility

---

1: input: learning rate $\eta > 0$
2: $\mathbf{x}_1 \leftarrow$ some point in $\mathcal{S}$
3: **for** $t = 1 \ldots T$ **do**
4:     play $\mathbf{x}_t$ and receive loss/payoff vector $\mathbf{f}_t \in \mathbb{R}^d$
5:     $\mathbf{y}_{t+1} \leftarrow \begin{cases} \mathbf{x}_t - \eta\mathbf{f}_t & \text{for losses} \\ \mathbf{x}_t + \eta\mathbf{f}_t & \text{for payoffs} \end{cases}$
6:     find $\mathbf{x}_{t+1} \in \mathbb{R}^d$ such that

$$\forall \mathbf{z} \in \mathcal{S} : \quad \|\mathbf{z} - \mathbf{x}_{t+1}\|^2 \leq \|\mathbf{z} - \mathbf{y}_{t+1}\|^2 \tag{3}$$

7: **end for**

---

**Lemma 2.2.** *[Online gradient descent without feasibility] Fix $\eta > 0$. Suppose Algorithm 1 is applied for $T$ rounds and let $\{\mathbf{f}_t\}_{t=1}^T \subset \mathbb{R}^d$ be the sequence of observed loss/payoff vectors, and let $\{\mathbf{x}_t\}_{t=1}^T$*

*be the sequence of points played by the algorithm. Then for any* $\mathbf{x} \in \mathcal{S}$ *it holds that*

$$\frac{1}{T} \sum_{t=1}^{T} \mathbf{x}_t \cdot \mathbf{f}_t - \frac{1}{T} \sum_{t=1}^{T} \mathbf{x} \cdot \mathbf{f}_t \leq \frac{1}{2T\eta} \|\mathbf{x}_1 - \mathbf{x}\|^2 + \frac{\eta}{2T} \sum_{t=1}^{T} \|\mathbf{f}_t\|^2 \quad \text{for losses;}$$

$$\frac{1}{T} \sum_{t=1}^{T} \mathbf{x} \cdot \mathbf{f}_t - \frac{1}{T} \sum_{t=1}^{T} \mathbf{x}_t \cdot \mathbf{f}_t \leq \frac{1}{2T\eta} \|\mathbf{x}_1 - \mathbf{x}\|^2 + \frac{\eta}{2T} \sum_{t=1}^{T} \|\mathbf{f}_t\|^2 \quad \text{for payoffs.}$$

The proof is given in the appendix for completeness.

## 3 Oracle-efficient Computation of (infeasible) Projections onto $\mathbf{CH}(\alpha\mathcal{K})$

In this section we detail our main technical tool for obtaining oracle-efficient online algorithms, i.e., our algorithm for computing projections, in the sense of Eq. (3), onto the convex set $\mathrm{CH}(\alpha\mathcal{K})$. Before presenting our projection algorithm, Algorithm 2 and detailing its theoretical guarantees, we first present the main algorithmic building block in the algorithm, which is described in the following lemma. Lemma 3.1 shows that for any point $\mathbf{x} \in \mathbb{R}^d$, we can either find a near-by point $\mathbf{p}$ which is a convex combination of points outputted by the extended approximation oracle (and hence, $\mathbf{p}$ is dominated by a convex combination of feasible points in $\mathcal{K}$ for any vector in $\mathcal{F}$, as discussed in Section 2.3.1), or we can find a separating hyperplane that separates $\mathbf{x}$ from $\mathrm{CH}(\alpha\mathcal{K})$ with sufficiently large margin. We achieve this by running the well known Ellipsoid method [10, 5] in a very specialized way. This application of the Ellipsoid method is similar in spirit to those in [15, 16], which applied this idea to computing *correlated equilibrium* in games and *algorithmic mechanism design*, though the implementation details and the way in which we apply this technique are quite different.

The proof of the following lemma is given in the appendix.

**Lemma 3.1** (Separation-or-Decomposition via the Ellipsoid method). *Fix* $\mathbf{x} \in \mathbb{R}^d$, $\epsilon \in (0, (\alpha+2)R]$, *and a positive integer* $N \geq cd^2 \ln\left(\frac{(\alpha+1)R+\|\mathbf{x}\|}{\epsilon}\right)$, *where* $c$ *is a positive universal constant. Consider an attempt to apply the Ellipsoid method for* $N$ *iterations to the following feasibility problem:*

$$\text{find } \mathbf{w} \in \mathbb{R}^d \text{ such that:} \qquad \forall \mathbf{z} \in \alpha\mathcal{K}: \quad (\mathbf{x} - \mathbf{z}) \cdot \mathbf{w} \geq \epsilon \quad \text{and} \quad \|\mathbf{w}\| \leq 1, \qquad (4)$$

*such that each iteration of the Ellipsoid method applies the following consecutive steps:*

1. $(\mathbf{v}, \mathbf{s}) \leftarrow \hat{\mathcal{O}}_{\mathcal{K}}(-\mathbf{w})$, *where* $\mathbf{w}$ *is the current iterate. If* $(\mathbf{x} - \mathbf{v}) \cdot \mathbf{w} < \epsilon$, *use* $\mathbf{v} - \mathbf{x}$ *as a separating hyperplane for the Ellipsoid method and continue to to the next iteration*

2. *if* $\|\mathbf{w}\| > 1$, *use* $\mathbf{w}$ *as a separating hyperplane for the Ellipsoid method and continue to the next iteration*

3. *otherwise* ($\|\mathbf{w}\| \leq 1$ *and* $(\mathbf{x} - \mathbf{v}) \cdot \mathbf{w} \geq \epsilon$), *declare Problem* (4) *feasible and return the vector* $\mathbf{w}$.

*Then, if the Ellipsoid method terminates declaring Problem 4 feasible, the returned vector* $\mathbf{w}$ *is a feasible solution to Problem* (4). *Otherwise (the Ellipsoid method completes* $N$ *iterations without declaring Problem* (4) *feasible), let* $(\mathbf{v}_1, \mathbf{s}_1), ..., (\mathbf{v}_N, \mathbf{s}_N)$ *be the outputs of the extended approximation oracle gathered throughout the run of the algorithm, and let* $(a_1, ..., a_N)$ *be an optimal solution to the following convex optimization problem:*

$$\min_{(a_1,...,a_N)} \frac{1}{2} \left\| \sum_{i=1}^{N} a_i \mathbf{v}_i - \mathbf{x} \right\|^2 \qquad \text{such that} \quad \forall i \in \{1, ..., N\}: a_i \geq 0, \quad \sum_{i=1}^{N} a_i = 1. \qquad (5)$$

*Then the point* $\mathbf{p} = \sum_{i=1}^{N} a_i \mathbf{v}_i$ *satisfies* $\|\mathbf{x} - \mathbf{p}\| \leq 3\epsilon$.

We are now ready to present our algorithm for computing projections onto $\mathrm{CH}(\alpha\mathcal{K})$ (in the sense of Eq. (3)). Consider now an attempt to project a point $\mathbf{y} \in \mathbb{R}^d$, and note that in particular, $\mathbf{y}$ itself is a valid projection (again, in the sense of Eq. (3)), however, in general, it is not a feasible point nor is it dominated by a convex combination of feasible points. When attempting to project $\mathbf{y} \in \mathbb{R}^d$, our algorithm continuously applies the *separation-or-decomposition* procedure described in Lemma 3.1.

In case the procedure returns a decomposition, then by Lemma 3.1, we have a point that is sufficiently close to $\mathbf{y}$ and is dominated for any vector in $\mathcal{F}$ by a convex combination (given explicitly) of feasible points in $\mathcal{K}$. Otherwise, the procedure returns a separating hyperplane which can be used to to "pull $\mathbf{y}$ closer" to $\text{CH}(\alpha\mathcal{K})$ in a way that the resulting point still satisfies the projection inequality given in Eq. (3), and the process then repeats itself. Since each time we obtain a hyperplane separating our current iterate from $\text{CH}(\alpha\mathcal{K})$, we pull the current iterate sufficiently towards $\text{CH}(\alpha\mathcal{K})$, this process must terminate. Lemma 3.2 gives exact bounds on the performance of the algorithm.

---

**Algorithm 2** (infeasible) Projection onto $\text{CH}(\alpha\mathcal{K})$

---

1: input: point $\mathbf{y} \in \mathbb{R}^d$, tolerance $\epsilon > 0$
2: $\tilde{\mathbf{y}} \leftarrow \mathbf{y}/\max\{1, \|\mathbf{y}\|/(\alpha R)\}$
3: **for** $t = 1 \ldots$ **do**
4:     call the SEPARATION-OR-DECOMPOSTION procedure (Lemma 3.1) with parameters $(\tilde{\mathbf{y}}, \epsilon)$
5:     **if** the procedure outputs a separating hyperplane $\mathbf{w}$ **then**
6:         $\tilde{\mathbf{y}} \leftarrow \tilde{\mathbf{y}} - \epsilon\mathbf{w}$
7:     **else**
8:         let $(a_1, ..., a_N), \{(\mathbf{v}_1, \mathbf{s}_1), ..., (\mathbf{v}_N, \mathbf{s}_N)\}$ be the decomposition returned
9:         **return** $\tilde{\mathbf{y}}, (a_1, ..., a_N), \{(\mathbf{v}_1, \mathbf{s}_1), ..., (\mathbf{v}_N, \mathbf{s}_N)\}$
10:     **end if**
11: **end for**

---

**Lemma 3.2.** *Fix $\mathbf{y} \in \mathbb{R}^d$ and $\epsilon \in (0, (\alpha+2)R]$. Algorithm 2 terminates after at most $\lceil \alpha^2 R^2/\epsilon^2 \rceil$ iterations, returning a point $\tilde{\mathbf{y}} \in \mathbb{R}^d$, a distribution $(a_1, ..., a_N)$ and a set $\{(\mathbf{v}_1, \mathbf{s}_1), ..., (\mathbf{v}_N, \mathbf{s}_N)\}$ outputted by the extended approximation oracle, where $N$ is as defined in Lemma 3.1, such that*

*1. $\forall \mathbf{z} \in CH(\alpha\mathcal{K}): \quad \|\tilde{\mathbf{y}} - \mathbf{z}\|^2 \leq \|\mathbf{y} - \mathbf{z}\|^2, \qquad$ 2. $\|\mathbf{p} - \tilde{\mathbf{y}}\| \leq 3\epsilon \quad for \quad \mathbf{p} := \sum_{i \in [N]} a_i \mathbf{v}_i.$*

*Moreover, if the **for** loop was entered a total number of $k$ times, then the final value of $\tilde{\mathbf{y}}$ satisfies*

$$dist^2(\tilde{\mathbf{y}}, CH(\alpha\mathcal{K})) \leq \min\{2\alpha^2 R^2, \ dist^2(\mathbf{y}, CH(\alpha\mathcal{K})) - (k-1)\epsilon^2\},$$

*and the overall number of queries to the approximation oracle is $O\left(kd^2 \ln((\alpha+1)R/\epsilon)\right)$.*

It is important to note that the worst case iteration bound in Lemma 3.2 does not seem so appealing for our purposes, since it depends polynomially on $1/\epsilon$, and in our online algorithms naturally we will need to take $\epsilon = O(T^{-c})$ for some $c > 0$, which seems to contradict our goal of achieving poly-logarithmic in $T$ oracle complexity, at least on average. However, as Lemma 3.2 shows, the more iterations Algorithm 2 performs, the closer it brings its final iterate to the set $\text{CH}(\alpha\mathcal{K})$. Thus, as we will show when analyzing the oracle complexity of our online algorithms, while a single call to Algorithm 2 can be expensive, when calling it sequentially, where each input is a small perturbation of the output of the previous call, the average number of iterations performed per such call cannot be too high.

# 4 Efficient Algorithms for the Full Information and Bandit Settings

We now turn to present our online algorithms for the full-information and bandit settings together with their regret bounds and oracle-complexity guarantees.

## 4.1 Algorithm for the full information setting

Our algorithm for the full-information setting, Algorithm 3, is given below.

**Theorem 4.1.** *[Main Theorem] Fix $\eta > 0, \epsilon \in (0, (\alpha+2)R]$. Suppose Algorithm 3 is applied for $T$ rounds and let $\{\mathbf{f}_t\}_{t=1}^T \subseteq \mathcal{F}$ be the sequence of observed loss/payoff vectors, and let $\{\mathbf{s}_t\}_{t=1}^T$ be the sequence of points played by the algorithm. Then it holds that*

$$\mathbb{E}\left[\alpha - regret\left(\{(\mathbf{s}_t, \mathbf{f}_t)\}_{t \in [T]}\right)\right] \leq \alpha^2 R^2 T^{-1}\eta^{-1} + \eta F^2/2 + 3F\epsilon,$$

*and the average number of calls to the approximation oracle of $\mathcal{K}$ per iteration is upper bounded by*

$$K(\eta, \epsilon) := O\left(\left(1 + \left(\eta\alpha RF + \eta^2 F^2\right)\epsilon^{-2}\right)d^2 \ln((\alpha+1)R/\epsilon)\right).$$

---

**Algorithm 3** Online Gradient Descent with Infeasible Projections onto $\text{CH}(\alpha\mathcal{K})$

---
1: input: learning rate $\eta > 0$, projection error parameter $\epsilon > 0$
2: $\mathbf{s}_1 \leftarrow$ some point in $\mathcal{K}$, $\tilde{\mathbf{y}}_1 \leftarrow \alpha\mathbf{s}_1$
3: **for** $t = 1\ldots T$ **do**
4: $\quad$ play $\mathbf{s}_t$ and receive loss/payoff vector $\mathbf{f}_t \in \mathcal{F}$
5: $\quad \mathbf{y}_{t+1} \leftarrow \begin{cases} \tilde{\mathbf{y}}_t - \eta\mathbf{f}_t & \text{if } \alpha \geq 1 \\ \tilde{\mathbf{y}}_t + \eta\mathbf{f}_t & \text{if } \alpha < 1 \end{cases}$
6: $\quad$ call Algorithm 2 with inputs $(\mathbf{y}_{t+1}, \epsilon)$ to obtain an approximated projection $\tilde{\mathbf{y}}_{t+1}$, a distribution $(a_1, ..., a_N)$ and $\{(\mathbf{v}_1, \mathbf{s}_1), ..., (\mathbf{v}_N, \mathbf{s}_N)\} \subseteq \mathbb{R}^d \times \mathcal{K}$, for some $N \in \mathbb{N}$.
7: $\quad$ sample $\mathbf{s}_{t+1} \in \{\mathbf{s}_1, ..., \mathbf{s}_N\}$ according to distribution $(a_1, ..., a_N)$
8: **end for**

---

*In particular, setting $\eta = \alpha R T^{-2/3}/F$, $\epsilon = \alpha R T^{-1/3}$ gives $\mathbb{E}\left[\alpha - regret\right] = O\left(\alpha R F T^{-1/3}\right)$, $K = O\left(d^2 \ln\left(\frac{\alpha+1}{\alpha}T\right)\right)$. Alternatively, setting $\eta = \alpha R T^{-1/2}/F$, $\epsilon = \alpha R T^{-1/2}$ gives $\mathbb{E}\left[\alpha - regret\right] = O\left(\alpha R F T^{-1/2}\right)$, $K = O\left(\sqrt{T}d^2 \ln\left(\frac{\alpha+1}{\alpha}T\right)\right)$.*

The proof is given in the appendix.

### 4.2 Algorithm for the bandit information setting

Our algorithm for the bandit setting follows from a very well known reduction from the bandit setting to the full information setting, also applied in the bandit algorithm of [13]. The algorithm simply simulates the full information algorithm, Algorithm 3, by providing it with estimated loss/payoff vectors $\hat{\mathbf{f}}_1, ..., \hat{\mathbf{f}}_T$ instead of the true vectors $\mathbf{f}_1, ..., \mathbf{f}_T$ which are not available in the bandit setting. This reduction is based on the use of a *Barycentric Spanner* (defined next) for the feasible set $\mathcal{K}$. As standard, we assume the points in $\mathcal{K}$ span the entire space $\mathbb{R}^d$, otherwise we can reformulate the problem in a lower-dimensional space, in which this assumption holds.

**Definition 4.1** (Barycentric Spanner[3])**.** *We say that a set of $d$ vectors $\{\mathbf{q}_1, ..., \mathbf{q}_d\} \subset \mathbb{R}^d$ is a Barycentric Spanner with parameter $\beta > 0$ for a set $\mathcal{S} \subset \mathbb{R}^d$, denoted by $\beta$-BS($\mathcal{S}$), if it holds that $\{\mathbf{q}_1, ..., \mathbf{q}_d\} \subset \mathcal{S}$, and the matrix $\mathbf{Q} := \sum_{i=1}^d \mathbf{q}_i \mathbf{q}_i^\top$ is not singular and $\max_{i \in [d]} \|\mathbf{Q}^{-1}\mathbf{q}_i\| \leq \beta$.*

Importantly, as discussed in [13], the assumption on the availability of such a set $\beta$-BS($\mathcal{K}$) seems reasonable, since i) for many sets that correspond to the set of all possible solutions to some well-studied NP-Hard optimization problem, one can still construct in poly$(d)$ time a barycentric spanner with $\beta = \text{poly}(d)$, ii) $\beta$-BS($\mathcal{K}$) needs to be constructed only once and then stored in memory (overall $d$ vectors in $\mathbb{R}^d$), and hence its construction can be viewed as a pre-processing step, and iii) as illustrated in [13], without further assumptions, the approximation oracle by itself is not sufficient to guarantee nontrivial regret bounds in the bandit setting.

The algorithm and the proof of the following theorem are given in the appendix.

**Theorem 4.2.** *Fix $\eta > 0, \epsilon \in (0, (\alpha + 2)R], \gamma \in (0, 1)$. Suppose Algorithm 5 is applied for $T$ rounds and let $\{\mathbf{f}_t\}_{t=1}^T \subseteq \mathcal{F}$ be the sequence of observed loss/payoff vectors, and let $\{\hat{\mathbf{s}}_t\}_{t=1}^T$ be the sequence of points played by the algorithm. Then it holds that*

$$\mathbb{E}\left[\alpha - regret\left(\{(\hat{\mathbf{s}}_t, \mathbf{f}_t)\}_{t \in [T]}\right)\right] \leq \alpha^2 R^2 \eta^{-1} T^{-1} + \eta d^2 C^2 \beta^2 \gamma^{-1}/2 + 3\epsilon F + \gamma C,$$

*and the expected number of calls to the approximation oracle of $\mathcal{K}$ per iteration is upper bounded by*

$$\mathbb{E}\left[K(\eta, \epsilon, \gamma)\right] := O\left(\left(1 + \left(\eta\alpha\beta dCR + (\eta dC\beta)^2/\gamma\right)\epsilon^{-2}\right)d^2 \ln\left((\alpha + 1)R/\epsilon\right)\right).$$

*In particular, setting $\eta = \frac{\alpha R}{\beta dC}T^{-2/3}$, $\epsilon = \alpha R T^{-1/3}$, $\gamma = T^{-1/3}$ gives $\mathbb{E}\left[\alpha - regret\right] = O\left((\alpha\beta dCR + \alpha RF + C)T^{-1/3}\right)$, $\mathbb{E}[K] = O\left(d^2 \ln\left(\frac{\alpha+1}{\alpha}T\right)\right)$.*

## Footnotes

[1]as we show in the appendix, even if we relax the algorithm of [13] to only guarantee $O(T^{-1/3})$ $\alpha$-regret, it will still require $O(T^{2/3})$ calls to the oracle per iteration, on average.

[2] we note that both of our assumptions that $\mathcal{K} \subset \mathbb{R}_+^d, \mathcal{F} \subset \mathbb{R}_+^d$ and that the oracle takes inputs from $\mathbb{R}_+^d$ are made for ease of presentation and clarity, and since these naturally hold for many NP-Hard optimization problem that are relevant to our setting. Nevertheless, these assumptions could be easily generalized as done in [13].

[3]this definition is somewhat different than the classical one given in [2], however it is equivalent to a $C$-approximate barycentric spanner [2], with an appropriately chosen constant $C(\beta)$.

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
