[Supplementary Material · paper_full.pdf]

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

# A The KKL approach

We now briefly describe how [13] use the extended approximation oracle and the online gradient descent without feasibility approach to construct their low $\alpha$-regret algorithm for the full information setting, and point out the limitation of this approach to obtaining low oracle complexity.

Consider some iteration $t$ of Algorithm 1 and let $\mathbf{y}_{t+1}$ be the newly computed point. Let $(\mathbf{x}, \mathbf{s}) \in \mathbb{R}^d \times \mathcal{K}$ be such that $\forall \mathbf{f} \in \mathcal{F} : \mathbf{x} \cdot \mathbf{f} \geq \mathbf{s} \cdot \mathbf{f}$ (e.g., take $\mathbf{x} = \mathbf{s}$), and let $(\mathbf{v}', \mathbf{s}') \leftarrow \hat{\mathcal{O}}_{\mathcal{K}}(\mathbf{x} - \mathbf{y}_{t+1})$. We have the following simple lemma.

**Lemma A.1.** *Fix $\epsilon \in (0, 3(\alpha+2)^2 R^2]$ and suppose that $\mathbf{x} \in \mathcal{B}(0, (\alpha+2)R)$. If $(\mathbf{x}-\mathbf{y}_{t+1})\cdot(\mathbf{x}-\mathbf{v}') \leq \epsilon$, then setting $\mathbf{x}_{t+1} \leftarrow \mathbf{x}$ gives*

$$\forall \mathbf{z} \in CH(\alpha\mathcal{K}) : \quad \|\mathbf{z} - \mathbf{x}_{t+1}\|^2 \leq \|\mathbf{z} - \mathbf{y}_{t+1}\|^2 + 2\epsilon.$$

*Otherwise, setting $\mathbf{x}' \leftarrow (1-\lambda)\mathbf{x} + \lambda\mathbf{v}'$, for appropriately chosen $\lambda \in (0,1)$, guarantees that*

$$\|\mathbf{x}' - \mathbf{y}_{t+1}\|^2 \leq \|\mathbf{x} - \mathbf{y}_{t+1}\|^2 - \Omega(\epsilon^2),$$

*and*

$$\forall \mathbf{f} \in \mathcal{F} : \qquad ((1-\lambda)\mathbf{s} + \lambda\mathbf{s}') \cdot \mathbf{f} \leq \mathbf{x}' \cdot \mathbf{f}.$$

*Proof.* To prove the first part of the lemma, suppose that $\mathbf{x}$ satisfies that $(\mathbf{x} - \mathbf{y}_{t+1}) \cdot (\mathbf{x} - \mathbf{v}') \leq \epsilon$, where $(\mathbf{v}', \mathbf{s}') \leftarrow \hat{\mathcal{O}}_{\mathcal{K}}(\mathbf{x} - \mathbf{y}_{t+1})$. Fix $\mathbf{z} \in \mathrm{CH}(\alpha\mathcal{K})$. It holds that

$$
\begin{aligned}
\|\mathbf{y}_{t+1} - \mathbf{z}\|^2 &= \|(\mathbf{y}_{t+1} - \mathbf{x}) + (\mathbf{x} - \mathbf{z})\|^2 = \|\mathbf{y}_{t+1} - \mathbf{x}\|^2 + \|\mathbf{x} - \mathbf{z}\|^2 - 2(\mathbf{x} - \mathbf{y}_{t+1}) \cdot (\mathbf{x} - \mathbf{z}) \\
&\geq \|\mathbf{x} - \mathbf{z}\|^2 - 2(\mathbf{x} - \mathbf{y}_{t+1}) \cdot (\mathbf{x} - \mathbf{z}) \\
&\geq \|\mathbf{x} - \mathbf{z}\|^2 - 2(\mathbf{x} - \mathbf{y}_{t+1}) \cdot (\mathbf{x} - \mathbf{v}') \geq \|\mathbf{x} - \mathbf{z}\|^2 - 2\epsilon,
\end{aligned}
$$

where the second inequality holds since $\mathbf{v}'$ is the output of the extended approximation oracle with respect to the input $(\mathbf{x} - \mathbf{y}_{t+1})$.

For the second part of the lemma, we observe that if $(\mathbf{x} - \mathbf{y}_{t+1}) \cdot (\mathbf{x} - \mathbf{v}') > \epsilon$, then

$$
\begin{aligned}
\|\mathbf{x}' - \mathbf{y}_{t+1}\|^2 &= \|\mathbf{x} - \mathbf{y}_{t+1} + \lambda(\mathbf{v}' - \mathbf{x})\|^2 \\
&= \|\mathbf{x} - \mathbf{y}_{t+1}\|^2 - 2\lambda(\mathbf{x} - \mathbf{y}_{t+1}) \cdot (\mathbf{x} - \mathbf{v}') + \lambda^2\|\mathbf{v}' - \mathbf{x}\|^2 \\
&\leq \|\mathbf{x} - \mathbf{y}_{t+1}\|^2 - 2\lambda\epsilon + 2\lambda^2(\|\mathbf{v}'\|^2 + \|\mathbf{x}\|^2) \\
&\leq \|\mathbf{x} - \mathbf{y}_{t+1}\|^2 - 2\lambda\epsilon + 4\lambda^2(\alpha+2)^2 R^2,
\end{aligned}
$$

where the first inequality follows since $(\mathbf{x}-\mathbf{y}_{t+1})\cdot(\mathbf{x}-\mathbf{v}') > \epsilon$ and using the triangle inequality with $(a+b)^2 \leq 2(a^2+b^2)$, and the second inequality follows by the assumption on $\mathbf{x}$ and since $\mathbf{v}'$ is the output of the extended approximation oracle. Thus, we can see that setting $\lambda = \frac{\epsilon}{3(\alpha+2)^2 R^2} \in (0,1]$, gives the requested result.

Finally, since $\mathbf{x}$ and $\mathbf{v}'$ are dominated by $\mathbf{s}$ and $\mathbf{s}'$ for any $\mathbf{f} \in \mathcal{F}$, respectively, we have that $\mathbf{x}' = (1-\lambda)\mathbf{x} + \lambda\mathbf{v}'$ is dominated by $(1-\lambda)\mathbf{s} + \lambda\mathbf{s}'$ for any $\mathbf{f} \in \mathcal{F}$. $\qquad\square$

Note that Lemma A.1 suggests an iterative algorithm to compute an $\epsilon$-approximated projection of $\mathbf{y}_{t+1}$ in Algorithm 1, that on each iteration reduces the potential $\|\mathbf{x} - \mathbf{y}_{t+1}\|^2$ by $\Omega(\epsilon^2)$, until finding an $\epsilon$-approximated projection of $\mathbf{y}_{t+1}$, $\mathbf{x}_{t+1}$, which must be found since the potential in non-negative. Moreover, this algorithm finds a point $\bar{\mathbf{s}}_{t+1} \in \mathrm{CH}(\mathcal{K})$, given explicitly as a convex combination of points in $\mathcal{K}$ (since $\lambda \in (0,1)$), such that $\bar{\mathbf{s}}_{t+1}$ dominates $\mathbf{x}_{t+1}$ for all vectors in $\mathcal{F}$. In particular, sampling $\mathbf{s}_{t+1}$ from this decomposition guarantees that we play a feasible point in $\mathcal{K}$, which in expectation, dominates $\mathbf{x}_{t+1}$ for all vectors in $\mathcal{F}$. The full algorithm, which is closely related to the classical Frank-Wolfe algorithm for convex optimization (a.k.a. the conditional gradient method) [5], is given below, see Algorithm 4[4].

**Lemma A.2.** *Fix $\epsilon \in (0, 3(\alpha+2)^2 R^2]$, $\eta > 0$ and a sequence of loss functions $\{\mathbf{f}_1, ..., \mathbf{f}_T\} \subseteq \mathcal{F}$. Consider the application of Algorithm 1 with learning rate $\eta$ when applied with respect to the feasible set $CH(\alpha\mathcal{K})$ and the sequence of losses $\{\mathbf{f}_1, ..., \mathbf{f}_T\} \subseteq \mathcal{F}$, and when we use the algorithm described*

*above to produce the (randomized) sequence of points $\{(\mathbf{x}_t, \mathbf{s}_t\}_{t\in[T]} \subset \mathbb{R}^d \times \mathcal{K}$. Then, focusing on the case $\alpha \geq 1$, it holds that,*

$$\mathbb{E}\left[\frac{1}{T}\sum_{t=1}^{T}\mathbf{s}_t \cdot \mathbf{f}_t\right] - \min_{\mathbf{x}\in CH(\alpha\mathcal{K})}\frac{1}{T}\sum_{t=1}^{T}\mathbf{x} \cdot \mathbf{f}_t = \mathbb{E}\left[\frac{1}{T}\sum_{t=1}^{T}\mathbf{s}_t \cdot \mathbf{f}_t\right] - \alpha\min_{\mathbf{x}\in\mathcal{K}}\frac{1}{T}\sum_{t=1}^{T}\mathbf{x} \cdot \mathbf{f}_t$$

$$\leq \frac{\alpha^2 R^2}{T\eta} + \frac{\eta F^2}{2} + \frac{\epsilon}{\eta}.$$

*Moreover, the number of calls to the extended approximation oracle per iteration $t$ is $O(\|\mathbf{y}_{t+1} - \mathbf{x}_t\|^2/\epsilon^2) = O(\eta^2 F^2/\epsilon^2)$, where the $O(\cdot)$ notation hides polynomial dependencies on $(1+\alpha), R$.*

*Proof.* We begin by proving the regret bound. Since each $\mathbf{x}_{t+1}$ is an approximated projection of $\mathbf{y}_{t+1}$ in the sense that

$$\forall \mathbf{z} \in \mathrm{CH}(\alpha\mathcal{K}): \quad \|\mathbf{z} - \mathbf{x}_{t+1}\|^2 \leq \|\mathbf{z} - \mathbf{y}_{t+1}\|^2 + 2\epsilon,$$

it is immediate to see from the proof of Lemma 2.2, that incorporating this approximation error into the regret bound, and bounding $\|\mathbf{f}_t\| \leq F$ for all $t$, results in the regret bound:

$$\frac{1}{T}\sum_{t=1}^{T}\mathbf{x}_t \cdot \mathbf{f}_t - \min_{\mathbf{x}\in CH(\alpha\mathcal{K})}\frac{1}{T}\sum_{t=1}^{T}\mathbf{x} \cdot \mathbf{f}_t = \frac{1}{T}\sum_{t=1}^{T}\mathbf{x}_t \cdot \mathbf{f}_t - \alpha\min_{\mathbf{x}\in\mathcal{K}}\frac{1}{T}\sum_{t=1}^{T}\mathbf{x} \cdot \mathbf{f}_t$$

$$\leq \frac{\alpha^2 R^2}{T\eta} + \frac{\eta F^2}{2} + \frac{\epsilon}{\eta}.$$

The regret bound now follows by recalling that for all $t$ and all $\mathbf{f} \in \mathcal{F}$: $\mathbb{E}[\mathbf{s}_t \cdot \mathbf{f}_t] = \bar{\mathbf{s}}_t \cdot \mathbf{f}_t \leq \mathbf{x}_t \cdot \mathbf{f}_t$, and taking expectation with respect to the randomness in $\mathbf{s}_t$.

To bound the number of calls to the approximation oracle per some iteration $t$, note that $\|\mathbf{x}_t - \mathbf{y}_{t+1}\|^2 \leq \eta^2 F^2$. Thus, if we initialize the projection algorithm, described in Lemma A.1, with the point $\mathbf{x}_t$, and we recall that each iteration of the algorithm reduces the potential $\|\mathbf{x} - \mathbf{y}_{t+1}\|^2$ by $\Omega(\epsilon^2)$, where $\mathbf{x}$ is the current iterate, then we have that at most $O(\eta^2 F^2/\epsilon^2)$ iterations are required for the algorithm to terminate. $\qquad\square$

The extra term of $\epsilon/\eta$ in the regret bound is due to fact we compute $\epsilon$-approximated projections.

It is clear that setting $\eta = O(1/\sqrt{T})$ and $\epsilon = O(1/T)$ in Lemma A.2 guarantees $O(T^{-1/2})$ expected $\alpha$-regret, which is optimal in $T$, however requires $O(T)$ calls to the approximation oracle per iteration. We can also observe that for any constants $a \in (0, 1), b \geq 1$, and sufficiently large $T$, Lemma A.2 cannot guarantee $O(T^{-a})$ expected $\alpha$-regret using only $O(\log^b T)$ calls to the approximation oracle per iteration, even on average. For this reason, in this paper we consider a drastically different algorithmic approach to applying the online gradient descent without feasibility methodology.

---

**Algorithm 4** Frank-Wolfe for Approximated (infeasible) Projection onto $\mathrm{CH}(\alpha\mathcal{K})$

---

1: input: point to project $\mathbf{y} \in \mathbb{R}^d$, error tolerance $\epsilon \in (0, \ 3(\alpha+2)^2 R^2)$
2: output: $(\mathbf{x}, \bar{\mathbf{s}}) \in \mathbb{R}^d \times \mathrm{CH}(\mathcal{K})$ such that $\mathbf{x}$ is an $\epsilon$-approximated infeasible projection of $\mathbf{y}$ dominated by $\bar{\mathbf{s}}$ for any $\mathbf{f} \in \mathcal{F}$
3: let $(\mathbf{x}_1, \bar{\mathbf{s}}_1) \in \mathbb{R}^n \times \mathrm{CH}(\mathcal{K})$ such that $\mathbf{x}_1$ is dominated by $\bar{\mathbf{s}}_1$ for any $\mathbf{f} \in \mathcal{F}$.
4: $\lambda \leftarrow \epsilon/(3(\alpha+2)^2 R^2)$
5: **for** $i = 1...$ **do**
6: $\quad (\mathbf{v}_i, \mathbf{s}_i) \leftarrow \hat{\mathcal{O}}_{\mathcal{K}}(\mathbf{x}_i - \mathbf{y})$
7: $\quad$ **if** $(\mathbf{x}_i - \mathbf{y}) \cdot (\mathbf{x}_i - \mathbf{v}_i) \leq \epsilon$ **then**
8: $\quad\quad$ **return** $(\mathbf{x}_i, \bar{\mathbf{s}}_i)$
9: $\quad$ **end if**
10: $\quad \mathbf{x}_{i+1} \leftarrow \mathbf{x}_i + \lambda(\mathbf{v}_i - \mathbf{x}_i)$
11: $\quad \bar{\mathbf{s}}_{i+1} \leftarrow \bar{\mathbf{s}}_i + \lambda(\mathbf{s}_i - \bar{\mathbf{s}}_i)$
12: **end for**

---

# B Proofs Omitted from Section 2

## B.1 Proof of Lamma 2.1

*Proof.* For the first item in the lemma note that for $\alpha \geq 1$ it holds that

$$
\begin{aligned}
\mathbf{v} \cdot \mathbf{c} &= \mathcal{O}_{\mathcal{K}}(\mathbf{c}^+) \cdot (\mathbf{c}^+ + \mathbf{c}^-) - \alpha R \bar{\mathbf{c}}^- \cdot (\mathbf{c}^+ + \mathbf{c}^-) \\
&\leq \alpha \min_{\mathbf{x} \in \mathcal{K}} \mathbf{x} \cdot \mathbf{c}^+ + \mathcal{O}_{\mathcal{K}}(\mathbf{c}^+) \cdot \mathbf{c}^- - \alpha R \|\mathbf{c}^-\| \\
&\leq \alpha \min_{\mathbf{x} \in \mathcal{K}} \mathbf{x} \cdot \mathbf{c}^+ - \alpha R \|\mathbf{c}^-\| \\
&\leq \alpha \min_{\mathbf{x} \in \mathcal{K}} \mathbf{x} \cdot \mathbf{c}^+ + \alpha \min_{\mathbf{x} \in \mathcal{K}} \mathbf{x} \cdot \mathbf{c}^- \\
&\leq \alpha \min_{\mathbf{x} \in \mathcal{K}} \mathbf{x} \cdot \mathbf{c} = \min_{\mathbf{x} \in \alpha\mathcal{K}} \mathbf{x} \cdot \mathbf{c},
\end{aligned}
$$

Similarly, for $\alpha > 1$ we have that

$$
\begin{aligned}
\mathbf{v} \cdot \mathbf{c} &= \mathcal{O}_{\mathcal{K}}(-\mathbf{c}^-) \cdot (\mathbf{c}^+ + \mathbf{c}^-) - R\bar{\mathbf{c}}^+ \cdot (\mathbf{c}^+ + \mathbf{c}^-) \\
&\leq -\alpha \max_{\mathbf{x} \in \mathcal{K}} \mathbf{x} \cdot (-\mathbf{c}^-) + \mathcal{O}_{\mathcal{K}}(-\mathbf{c}^-) \cdot \mathbf{c}^+ - R\|\mathbf{c}^+\| \\
&\leq \alpha \min_{\mathbf{x} \in \mathcal{K}} \mathbf{x} \cdot \mathbf{c}^- + R\|\mathbf{c}^+\| - R\|\mathbf{c}^+\| \\
&\leq \alpha \min_{\mathbf{x} \in \mathcal{K}} \mathbf{x} \cdot \mathbf{c}^- + \alpha \min_{\mathbf{x} \in \mathcal{K}} \mathbf{x} \cdot \mathbf{c}^+ \\
&\leq \alpha \min_{\mathbf{x} \in \mathcal{K}} \mathbf{x} \cdot \mathbf{c} = \min_{\mathbf{x} \in \alpha\mathcal{K}} \mathbf{x} \cdot \mathbf{c}.
\end{aligned}
$$

For the second item, it suffices to observe that for $\alpha \geq 1$ we have that $\mathbf{s} \leq \mathbf{v}$ (coordinate-wise) and hence for every $\mathbf{f} \in \mathcal{F}$ we have that $\mathbf{s} \cdot \mathbf{f} \leq \mathbf{v} \cdot \mathbf{f}$ (recall that $\mathcal{F} \subset \mathbb{R}_+^d$). Similarly, when $\alpha < 1$, we note that $\mathbf{s} \geq \mathbf{v}$.

The third item holds trivially. $\qquad\square$

## B.2 Proof of Lamma 2.2

*Proof.* Fix $\mathbf{x} \in \mathcal{S}$. Assume that the vectors $\mathbf{f}_1, ..., \mathbf{f}_T$ are losses. By the definition of the infeasible projection $\mathbf{x}_{t+1}$, for any iteration $t \geq 1$ it holds that

$$
\begin{aligned}
\|\mathbf{x}_{t+1} - \mathbf{x}\|^2 &\leq \|\mathbf{y}_{t+1} - \mathbf{x}\|^2 = \|\mathbf{x}_t - \eta\mathbf{f}_t - \mathbf{x}\|^2 \\
&= \|\mathbf{x}_t - \mathbf{x}\|^2 - 2\eta(\mathbf{x}_t - \mathbf{x}) \cdot \mathbf{f}_t + \eta^2 \|\mathbf{f}_t\|^2
\end{aligned}
$$

Rearranging and summing over all iterations we have that

$$
\begin{aligned}
\sum_{t=1}^{T} (\mathbf{x}_t - \mathbf{x}) \cdot \mathbf{f}_t &\leq \frac{1}{2\eta} \sum_{t=1}^{T} (\|\mathbf{x}_t - \mathbf{x}\|^2 - \|\mathbf{x}_{t+1} - \mathbf{x}\|^2) + \frac{\eta}{2} \sum_{t=1}^{T} \|\mathbf{f}_t\|^2 \\
&\leq \frac{1}{2\eta} \|\mathbf{x}_1 - \mathbf{x}\|^2 + \frac{\eta}{2} \sum_{t=1}^{T} \|\mathbf{f}_t\|^2.
\end{aligned}
$$

It is immediate to see that the proof of the result in case of payoffs instead of losses (for which the only change is in the update of $\mathbf{y}_{t+1}$ in Algorithm 1), follows the same lines as the one for losses given above. $\qquad\square$

# C Lemmas and Proofs Omitted from Section 3

## C.1 Proof of Lemma 3.1

*Proof.* To prove the first part of the lemma, suppose there exists some iteration during which the Ellipsoid method declares Problem (4) feasible, and let $\mathbf{w}$ be the corresponding iterate and let $(\mathbf{v}, \mathbf{s})$ be the output of the extended approximation oracle on that iteration. Clearly it holds that

$$
\epsilon \leq (\mathbf{x} - \mathbf{v}) \cdot \mathbf{w} = \mathbf{x} \cdot \mathbf{w} + \mathbf{v} \cdot (-\mathbf{w}) \leq \mathbf{x} \cdot \mathbf{w} + \min_{\mathbf{z} \in \alpha\mathcal{K}} \mathbf{z} \cdot (-\mathbf{w}) = \min_{\mathbf{z} \in \alpha\mathcal{K}} (\mathbf{x} - \mathbf{z}) \cdot \mathbf{w},
$$

where the first inequality follows from the fact that the Ellipsoid method declared Problem (4) feasible, and the second inequality follows from the definition of the extended approximation oracle. Since the Ellipsoid method declared Problem (4) feasible, it also follows that $\|\mathbf{w}\| \leq 1$ and hence $\mathbf{w}$ is indeed a feasible solution to Problem (4).

Consider now the case that all $N$ iterations are executed without declaring Problem (4) feasible and let $\mathbf{v}_1, ..., \mathbf{v}_N$ be as defined in the lemma. We would like to show that this implies that

$$\forall \text{ unit vector } \mathbf{w}: \quad \min_{i \in \{1,...,N'\}} (\mathbf{x} - \mathbf{v}_i) \cdot \mathbf{w} \leq 3\epsilon. \tag{6}$$

Then, the second part of the lemma follows from applying the next lemma, Lemma C.1, which shows that (6) implies that the point $\mathbf{p}$ defined in the lemma indeed satisfies $\|\mathbf{p} - \mathbf{x}\| \leq 3\epsilon$, as required.

Towards proving (6), suppose that there exists a unit vector $\mathbf{h} \in \mathbb{R}^d$ such that for all $i \in \{1, ..., N\}$, $(\mathbf{x} - \mathbf{v}_i) \cdot \mathbf{h} > 3\epsilon$. It follow that $\forall i \in \{1, ..., N\} : (\mathbf{x} - \mathbf{v}_i) \cdot \mathbf{h}/2 > 3\epsilon/2$. It follows from a simple application of the Cauchy-Swartz inequality and the observation that $\|\mathbf{x} - \mathbf{v}_i\| \leq \|\mathbf{x}\| + \|\mathbf{v}_i\| \leq \|\mathbf{x}\| + (\alpha + 2)R$, that denoting $r := \frac{\epsilon}{2(\alpha+2)R + \|\mathbf{x}\|}$, we have that

$$\forall \mathbf{h}' \in \mathcal{B}(\mathbf{h}/2, r) : \quad \min_{i \in [N]} (\mathbf{x} - \mathbf{v}_i) \cdot \mathbf{h}' > \epsilon. \tag{7}$$

Note that on one hand, by the above and our assumption on $\epsilon$, every point in $\mathcal{B}(\mathbf{h}/2, r)$ satisfies the stopping criteria of the Ellipsoid method described in the lemma. On the other-hand, on every iteration in which the current iterate $\mathbf{w}$ is not declared feasible, it follows that the separating hyperplane fed to the Ellipsoid method indeed separates $\mathbf{w}$ from $\mathcal{B}(\mathbf{h}/2, r)$. To see why this is true, we consider the two possible options for the separating hyperplane. If the hyperplane is $\mathbf{v}_i - \mathbf{x}$, where $\mathbf{v}_i$ is the output of the extended approximation oracle on that iteration, then we have that

$$\forall \mathbf{h}' \in \mathcal{B}(\mathbf{h}/2, r) : \quad (\mathbf{w} - \mathbf{h}') \cdot (\mathbf{v}_i - \mathbf{x}) = (\mathbf{x} - \mathbf{v}_i) \cdot \mathbf{h}' - (\mathbf{x} - \mathbf{v}_i) \cdot \mathbf{w} > \epsilon - \epsilon = 0,$$

where the first inequality follows from Eq. (7) and the fact that $(\mathbf{x} - \mathbf{v}_i) \cdot \mathbf{w} < \epsilon$ on this iteration. If the hyperplane used was $\mathbf{w}$, which guarantees that on that iteration $\|\mathbf{w}\| > 1$, then we have that

$$\forall \mathbf{h}' \in \mathcal{B}(\mathbf{h}/2, r) : \quad (\mathbf{w} - \mathbf{h}') \cdot \mathbf{w} = \|\mathbf{w}\| - \mathbf{w} \cdot \mathbf{h}' > 1 - 1 = 0,$$

where the last inequality follows since by our assumption on $\epsilon$, it holds that $\mathcal{B}(\mathbf{h}/2, r) \subset \mathcal{B}(0, 1)$. Thus, we can conclude that if the number of Ellipsoid method iterations satisfies $N \geq cd^2 \ln\left(\frac{(\alpha+1)R + \|\mathbf{x}\|}{\epsilon}\right)$ for an appropriate universal constant $c > 0$, and all $N$ iterations were completed without declaring feasibility, it follows that no such unit vector $\mathbf{h}$ can exist, which means Eq. (6) holds, and the result follows. $\square$

**Lemma C.1.** *Fix $\mathbf{x} \in \mathbb{R}^d$, vectors $\mathbf{v}_1, ..., \mathbf{v}_N \in \mathbb{R}^d$ and $\epsilon > 0$. If for any unit vector $\mathbf{w}$ it holds that $\min_{i \in \{1,...,N\}} (\mathbf{x} - \mathbf{v}_i) \cdot \mathbf{w} \leq \epsilon$, then it follows that the point $\mathbf{p} = \sum_{i=1}^N a_i \mathbf{v}_i$, where $(a_1, ..., a_N)$ is an optimal solution to Problem (5), satisfies $\|\mathbf{p} - \mathbf{x}\| \leq \epsilon$.*

*Proof.* First we show that the following holds:

$$\begin{aligned} \forall i, j \text{ s.t. } a_i > 0, a_j > 0 : \quad & (\mathbf{p} - \mathbf{x}) \cdot \mathbf{v}_i = (\mathbf{p} - \mathbf{x}) \cdot \mathbf{v}_j, \\ \forall i, j \text{ s.t. } a_i > 0, a_j = 0 : \quad & (\mathbf{p} - \mathbf{x}) \cdot \mathbf{v}_i \leq (\mathbf{p} - \mathbf{x}) \cdot \mathbf{v}_j. \end{aligned} \tag{8}$$

To see why this is true, fix some $i, j$ such that $a_i > 0$ and consider the point $\mathbf{p}' = \mathbf{p} + \delta(\mathbf{v}_j - \mathbf{v}_i)$ such that $0 < \delta \leq a_i$. Clearly $\mathbf{p}'$ lies in the convex hull of $\{\mathbf{v}_1, ..., \mathbf{v}_N\}$ and hence is a feasible solution to Problem (5). It holds that

$$\frac{1}{2}\|\mathbf{p}' - \mathbf{x}\|^2 = \frac{1}{2}\|\mathbf{p} - \mathbf{x}\|^2 + \delta(\mathbf{v}_j - \mathbf{v}_i) \cdot (\mathbf{p} - \mathbf{x}) + \frac{\delta^2}{2}\|\mathbf{v}_i - \mathbf{v}_j\|^2. \tag{9}$$

Thus, we can see that if (8) does not hold, then without loss of generality we can always choose $i, j$ such that $a_i > 0$ and $(\mathbf{p} - \mathbf{x}) \cdot \mathbf{v}_i > (\mathbf{p} - \mathbf{x}) \cdot \mathbf{v}_j$, and thus as can be seen from Eq. (9), choosing $\delta$ to be sufficiently small it follows that $\|\mathbf{p}' - \mathbf{x}\|^2 < \|\mathbf{p} - \mathbf{x}\|^2$, contradicting the optimality of $\mathbf{p}$.

Denoting by $\mathbf{u}$ the unit vector in the direction of $\mathbf{x} - \mathbf{p}$, we can rewrite Eq. (8) as follows:

$$\begin{aligned} \forall i, j \text{ s.t. } a_i > 0, a_j > 0 : \quad & (\mathbf{x} - \mathbf{v}_i) \cdot \mathbf{u} = (\mathbf{x} - \mathbf{v}_j) \cdot \mathbf{u}, \\ \forall i, j \text{ s.t. } a_i > 0, a_j = 0 : \quad & (\mathbf{x} - \mathbf{v}_i) \cdot \mathbf{u} \leq (\mathbf{x} - \mathbf{v}_j) \cdot \mathbf{u}. \end{aligned} \tag{10}$$

Using our assumption, we in particular have that $\min_{i \in [N]}(\mathbf{x} - \mathbf{v}_i) \cdot \mathbf{u} \leq \epsilon$, and using Eq. (10) we have that

$$\|\mathbf{p} - \mathbf{x}\| = (\mathbf{x} - \mathbf{p}) \cdot \mathbf{u} = \sum_{i=1}^{N} a_i(\mathbf{x} - \mathbf{v}_i) \cdot \mathbf{u} = \min_{i \in [N]}(\mathbf{x} - \mathbf{v}_i) \cdot \mathbf{u} \leq \epsilon,$$

where the last equality is a consequence of Eq. (10) and the fact that $(a_1, ..., a_N)$ is a distribution. Thus the lemma follows. $\qquad\square$

### C.2 Proof of Lemma 3.2

*Proof.* Note that the second item in the lemma is a straightforward guarantee of Lemma 3.1.

To prove the first item, suppose that the algorithm terminates after the **for** loop was entered $k$ times, and let $\tilde{\mathbf{y}}_1, ..., \tilde{\mathbf{y}}_k$ denote the values of $\tilde{\mathbf{y}}$ throughout the run of the algorithm, where $\tilde{\mathbf{y}}_i$ is the value of $\tilde{\mathbf{y}}$ at the beginning of the $i$th iteration of the **for** loop. Note that since $\mathrm{CH}(\alpha\mathcal{K}) \subseteq \mathcal{B}(0, \alpha R)$ and $\tilde{\mathbf{y}}_1$ is the projection of $\mathbf{y}$ onto $\mathcal{B}(0, \alpha R)$, we have that $\forall \mathbf{z} \in \mathrm{CH}(\alpha\mathcal{K}) : \|\tilde{\mathbf{y}}_1 - \mathbf{z}\|^2 \leq \|\mathbf{y} - \mathbf{z}\|^2$.

We are now going to show that for any $i \geq 1$ it holds that

$$\forall \mathbf{z} \in \mathrm{CH}(\alpha\mathcal{K}) : \quad \|\tilde{\mathbf{y}}_{i+1} - \mathbf{z}\|^2 \leq \|\tilde{\mathbf{y}}_i - \mathbf{z}\|^2, \tag{11}$$

which clearly yields item 1 in the lemma.

To prove that Eq. (11) holds throughout the run of the algorithm, consider an iteration $i$ of the **for** loop during which, the SEPARATION-OR-DECOMPOSTION procedure returns a separating hyperplane $\mathbf{w}$. It holds that

$$
\begin{aligned}
\forall \mathbf{z} \in \mathrm{CH}(\alpha\mathcal{K}) : \quad \|\tilde{\mathbf{y}}_i - \mathbf{z}\|^2 &= \|\tilde{\mathbf{y}}_i - \tilde{\mathbf{y}}_{i+1} + \tilde{\mathbf{y}}_{i+1} - \mathbf{z}\|^2 \\
&= \|\tilde{\mathbf{y}}_i - \tilde{\mathbf{y}}_{i+1}\|^2 + \|\tilde{\mathbf{y}}_{i+1} - \mathbf{z}\|^2 + 2(\tilde{\mathbf{y}}_i - \tilde{\mathbf{y}}_{i+1}) \cdot (\tilde{\mathbf{y}}_{i+1} - \mathbf{z}) \\
&\geq \|\tilde{\mathbf{y}}_{i+1} - \mathbf{z}\|^2 + 2(\tilde{\mathbf{y}}_i - \tilde{\mathbf{y}}_{i+1}) \cdot (\tilde{\mathbf{y}}_{i+1} - \mathbf{z}) \\
&= \|\tilde{\mathbf{y}}_{i+1} - \mathbf{z}\|^2 + 2\epsilon\mathbf{w} \cdot [(\tilde{\mathbf{y}}_i - \mathbf{z}) - \epsilon\mathbf{w}] \\
&= \|\tilde{\mathbf{y}}_{i+1} - \mathbf{z}\|^2 + 2\epsilon(\tilde{\mathbf{y}}_i - \mathbf{z}) \cdot \mathbf{w} - 2\epsilon^2\|\mathbf{w}\|^2 \\
&\geq \|\tilde{\mathbf{y}}_{i+1} - \mathbf{z}\|^2,
\end{aligned}
$$

where the third equality follows from the update rule of $\tilde{\mathbf{y}}$ in the algorithm, and the last inequality is a direct consequence of the guarantees of Lemma 3.1. Thus, Eq. (11) indeed holds for all $i \geq 1$, which gives the first item listed in the lemma.

We now turn to upper bound the number of iterations performed by the algorithm. Consider again an iteration $i$ of the loop during which the SEPARATION-OR-DECOMPOSTION procedure returns a separating hyperplane $\mathbf{w}$. We are going to show that

$$\mathrm{dist}^2(\tilde{\mathbf{y}}_{i+1}, \mathrm{CH}(\alpha\mathcal{K})) \leq \mathrm{dist}^2(\tilde{\mathbf{y}}_i, \mathrm{CH}(\alpha\mathcal{K})) - \epsilon^2,$$

which, together with the fact that $\mathrm{dist}^2(\tilde{\mathbf{y}}_1, \mathrm{CH}(\alpha\mathcal{K})) \leq 2\alpha^2 R^2$, gives the desired upper bound on the number of iterations.

Denote $\mathbf{x}_i = \arg\min_{\mathbf{x} \in \mathrm{CH}(\alpha\mathcal{K})} \|\mathbf{x} - \tilde{\mathbf{y}}_i\|$ and $\mathbf{x}_{i+1} = \arg\min_{\mathbf{x} \in \mathrm{CH}(\alpha\mathcal{K})} \|\mathbf{x} - \tilde{\mathbf{y}}_{i+1}\|$. It holds that

$$
\begin{aligned}
\mathrm{dist}^2(\tilde{\mathbf{y}}_{i+1}, \mathrm{CH}(\alpha\mathcal{K})) &= \|\mathbf{x}_{i+1} - \tilde{\mathbf{y}}_{i+1}\|^2 \leq \|\mathbf{x}_i - \tilde{\mathbf{y}}_{i+1}\|^2 = \|\mathbf{x}_i - \tilde{\mathbf{y}}_i + \epsilon\mathbf{w}\|^2 \\
&= \mathrm{dist}^2(\tilde{\mathbf{y}}_i, \mathrm{CH}(\alpha\mathcal{K})) + \epsilon^2\|\mathbf{w}\|^2 - 2\epsilon(\tilde{\mathbf{y}}_i - \mathbf{x}_i) \cdot \mathbf{w} \\
&\leq \mathrm{dist}^2(\tilde{\mathbf{y}}_i, \mathrm{CH}(\alpha\mathcal{K})) - \epsilon^2,
\end{aligned}
$$

where the inequality is a direct consequence of the guarantees of Lemma 3.1. Thus, we obtain both the desired bound on the number of iterations and the bound on the distance of the final point $\tilde{\mathbf{y}}$ from $\mathrm{CH}(\alpha\mathcal{K})$.

Finally, we turn to upper bound to overall number of queries to the approximation oracle. Using the bound in Lemma 3.1, we have that the number of calls to the oracle on the $i$th iteration of the loop is

upper bounded by $O\left(d^2 \ln\left(\frac{(\alpha+1)R+\|\tilde{\mathbf{y}}_i\|}{\epsilon}\right)\right)$. As we have shown, the values $\mathrm{dist}(\tilde{\mathbf{y}}_i, \mathrm{CH}(\alpha\mathcal{K}))$ are monotonically decreasing with $i$ and hence we can upper bound

$$\|\tilde{\mathbf{y}}_i\| \leq \max_{\mathbf{x}\in\mathrm{CH}(\alpha\mathcal{K})} \|\mathbf{x}\| + \mathrm{dist}(\tilde{\mathbf{y}}_i, \mathrm{CH}(\alpha\mathcal{K})) \leq \alpha R + \mathrm{dist}(\tilde{\mathbf{y}}_1, \mathrm{CH}(\alpha\mathcal{K})) \leq \alpha R + \sqrt{2}\alpha R,$$

where the last inequality holds since $\tilde{\mathbf{y}}_1$ is the projection of $\mathbf{y}$ onto the ball $\mathcal{B}(0, \alpha R)$. Thus, the overall number of queries to the approximation oracle after $k$ iterations is upper bounded by $O\left(kd^2 \ln\left(\frac{(\alpha+1)R}{\epsilon}\right)\right)$. □

# D  Algorithms and Proofs Omitted from Section 4

## D.1  Proof of Theorem 4.1

*Proof.* For the proof we focus on the case $\alpha \geq 1$ since the proof for the complementary follows from the same derivations up to changes in the obvious places. To prove the regret bound, we simply apply Lemma 2.2 with respect to the sequence of points $\{\tilde{\mathbf{y}}_t\}_{t=1}^T$ and the feasible set $\mathrm{CH}(\alpha\mathcal{K})$ and plugin the guarantee of Lemma 3.2, which gives

$$
\begin{aligned}
\sum_{t=1}^T \tilde{\mathbf{y}}_t \cdot \mathbf{f}_t - \min_{\mathbf{x}\in\alpha\mathcal{K}} \sum_{t=1}^T \mathbf{x} \cdot \mathbf{f}_t &= \sum_{t=1}^T \tilde{\mathbf{y}}_t \cdot \mathbf{f}_t - \alpha \cdot \min_{\mathbf{x}\in\mathcal{K}} \sum_{t=1}^T \mathbf{x} \cdot \mathbf{f}_t \\
&\leq \frac{\alpha^2 R^2}{\eta} + T\frac{\eta F^2}{2},
\end{aligned}
$$

where we have used the the fact that $\|\tilde{\mathbf{y}}_1\| \leq \alpha R$ and $\|\mathbf{f}_t\| \leq F$ for all $t \in [T]$. For every iteration $t \geq 1$, let us denote $\mathbf{p}_{t+1} = \sum_{i=1}^N a_i\mathbf{v}_i$, $\bar{\mathbf{s}}_t = \sum_{i=1}^N a_i\mathbf{s}_i$, where $(a_1, ..., a_N)$, $\{(\mathbf{v}_1, \mathbf{s}_1), ..., (\mathbf{v}_N, \mathbf{s}_N)\}$ are the outputs of the call to Algorithm 2 on iteration $t$, and for $t = 1$ we denote $\mathbf{p}_1 = \tilde{\mathbf{y}}_1$ and $\bar{\mathbf{s}}_1 = \mathbf{s}_1$. By the guarantee of Lemma 3.2, we have that

$$\sum_{t=1}^T \mathbf{p}_t \cdot \mathbf{f}_t - \alpha \cdot \min_{\mathbf{x}\in\mathcal{K}} \sum_{t=1}^T \mathbf{x} \cdot \mathbf{f}_t \leq \frac{\alpha^2 R^2}{\eta} + T\frac{\eta F^2}{2} + 3T\epsilon F,$$

where the inequality holds since for all $t \geq 1$: $|(\mathbf{p}_t - \tilde{\mathbf{y}}_t) \cdot \mathbf{f}_t| \leq \|\mathbf{p}_t - \tilde{\mathbf{y}}_t\| \cdot \|\mathbf{f}_t\| \leq 3\epsilon F$. The regret bound now follows since for any iteration $t$, $\bar{\mathbf{s}}_t$ dominates $\mathbf{p}_t$ for any vector $\mathbf{f} \in \mathcal{F}$, and since $\mathbb{E}[\mathbf{s}_t] = \bar{\mathbf{s}}_t$.

We now turn to upper bound the overall number of queries to the approximation oracle of $\mathcal{K}$. Let $k_t$ be the number of iterations it took Algorithm 2 to terminate, when invoked on iteration $t$ of Algorithm 3. Note that, by Lemma 3.2, we have that $K(\eta, \epsilon) = O\left(\frac{1}{T}\sum_{t=1}^{T-1} k_t d^2 \ln\left(\frac{(\alpha+1)R}{\epsilon}\right)\right)$. By Lemma 3.2, it follows that on any iteration $t$,

$$
\begin{aligned}
\mathrm{dist}^2(\tilde{\mathbf{y}}_{t+1}, \mathrm{CH}(\alpha\mathcal{K})) &\leq \mathrm{dist}^2(\mathbf{y}_{t+1}, \mathrm{CH}(\alpha\mathcal{K})) - (k_t - 1)\epsilon^2 \\
&= \mathrm{dist}^2(\tilde{\mathbf{y}}_t - \eta\mathbf{f}_t, \mathrm{CH}(\alpha\mathcal{K})) - (k_t - 1)\epsilon^2 \\
&\leq (\mathrm{dist}(\tilde{\mathbf{y}}_t, \mathrm{CH}(\alpha\mathcal{K})) + \eta F)^2 - (k_t - 1)\epsilon^2 \\
&= \mathrm{dist}^2(\tilde{\mathbf{y}}_t, \mathrm{CH}(\alpha\mathcal{K})) + 2\eta F\mathrm{dist}(\tilde{\mathbf{y}}_t, \mathrm{CH}(\alpha\mathcal{K})) + \eta^2 F^2 - k_t\epsilon^2 + \epsilon^2.
\end{aligned}
$$

Rearranging, summing over all $T$ iterations, and recalling that for all $t$, $\mathrm{dist}(\tilde{\mathbf{y}}_t, \mathrm{CH}(\alpha\mathcal{K})) \leq \sqrt{2}\alpha R$, we have that

$$
\begin{aligned}
\sum_{t=1}^{T-1} k_t &\leq \frac{1}{\epsilon^2}\left(\mathrm{dist}^2(\tilde{\mathbf{y}}_1, \mathrm{CH}(\alpha\mathcal{K})) - \mathrm{dist}^2(\tilde{\mathbf{y}}_T, \mathrm{CH}(\alpha\mathcal{K})) + (T-1)\left(2\sqrt{2}\eta\alpha RF + \eta^2 F^2 + \epsilon^2\right)\right) \\
&\leq (T-1)\left(1 + \frac{2\sqrt{2}\eta\alpha RF + \eta^2 F^2}{\epsilon^2}\right).
\end{aligned}
$$

□

## D.2 Algorithm for the bandit setting and proof of Theorem 4.2

---

**Algorithm 5** Bandit Algorithm

---

1: input: learning rate $\eta > 0$, projection error parameter $\epsilon > 0$, $\{\mathbf{q}_1, ..., \mathbf{q}_d\}$ - a $\beta$-BS$(\mathcal{K})$ for some $\beta > 0$, exploration parameter $\gamma \in (0, 1)$
2: instantiate Algorithm 3 with parameters $(\eta, \epsilon)$
3: **for** $t = 1 \dots T$ **do**
4:     receive $(\mathbf{s}_t, \tilde{\mathbf{y}}_t) \in \mathcal{K} \times \mathcal{B}(0, \alpha R)$ from Algorithm 3
5:     $b_t \leftarrow \begin{cases} \text{EXPLORE} & \text{with prob. } \gamma \\ \text{EXPLOIT} & \text{with prob. } 1 - \gamma \end{cases}$
6:     **if** $b_t = \text{EXPLORE}$ **then**
7:        sample $i_t \in [d]$ uniformly at random
8:        play $\hat{\mathbf{s}}_t = \mathbf{q}_{i_t}$
9:        receive loss/payoff $\ell_t = \mathbf{q}_i \cdot \mathbf{f}_t$
10:       set $\hat{\mathbf{f}}_t \leftarrow \frac{d\ell_t}{\gamma} \mathbf{Q}^{-1} \mathbf{q}_{i_t}$ {recall $\mathbf{Q} = \sum_{i=1}^{d} \mathbf{q}_i \mathbf{q}_i^{\top}$}
11:     **else**
12:        play $\hat{\mathbf{s}}_t = \mathbf{s}_t$
13:        receive loss/payoff $\ell_t = \mathbf{s}_t \cdot \mathbf{f}_t$
14:        set $\hat{\mathbf{f}}_t \leftarrow \mathbf{0}$
15:     **end if**
16:     feed $\hat{\mathbf{f}}_t$ to Algorithm 3 as the loss/payoff vector for round $t$
17: **end for**

---

*proof of Theorem 4.2.* The proof is very similar to that of Theorem 4.1 and we focus on the modifications of it required to prove Theorem 4.2. Again, we focus on the case $\alpha \geq 1$ since the complementary case follows the same lines with the obvious modifications. Let $\mathbf{x}^* \in \arg\min_{\mathbf{x} \in \mathcal{K}} \sum_{t=1}^{T} \mathbf{x} \cdot \mathbf{f}_t$. Applying Lemma 2.2 with respect to the sequence of points $\{\tilde{\mathbf{y}}_t\}_{t=1}^{T}$ and the sequence of losses $\{\hat{\mathbf{f}}_t\}_{t=1}^{T}$, we have that

$$\sum_{t=1}^{T} \tilde{\mathbf{y}}_t \cdot \hat{\mathbf{f}}_t - \alpha \cdot \sum_{t=1}^{T} \mathbf{x}^* \cdot \hat{\mathbf{f}}_t \leq \frac{\alpha^2 R^2}{\eta} + \frac{\eta}{2} \sum_{t=1}^{T} \|\hat{\mathbf{f}}_t\|^2.$$

Taking expectation with respect to the random variables $b_1, i_1, ..., b_T, i_T$ and noting that for all $t \in [T]$, both $\mathbf{x}^*$ and $\tilde{\mathbf{y}}_t$ are independent of the randomness in $\hat{\mathbf{f}}_t$, we have that

$$\mathbb{E}_{\{(b_t, i_t)\}_{t=1}^{T}} \left[ \sum_{t=1}^{T} \tilde{\mathbf{y}}_t \cdot \mathbf{f}_t \right] - \alpha \cdot \sum_{t=1}^{T} \mathbf{x}^* \cdot \mathbf{f}_t \leq \frac{\alpha^2 R^2}{\eta} + T \frac{\eta d^2 C^2 \beta^2}{2\gamma},$$

were we have used the observations that

$$\mathbb{E}_{b_t, i_t}[\hat{\mathbf{f}}_t] = \gamma \sum_{i=1}^{d} \frac{1}{d} \cdot \frac{d\mathbf{q}_i^{\top} \mathbf{f}_t}{\gamma} \mathbf{Q}^{-1} \mathbf{q}_i = \sum_{i=1}^{d} \mathbf{Q}^{-1} \mathbf{q}_i \mathbf{q}_i^{\top} \mathbf{f}_t = \mathbf{Q}^{-1} \mathbf{Q} \mathbf{f}_t = \mathbf{f}_t,$$

$$\mathbb{E}_{b_t}[\|\hat{\mathbf{f}}_t\|^2] = \gamma \frac{d^2}{\gamma^2} \ell_t^2 \|\mathbf{Q}^{-1} \mathbf{q}_{i_t}\|^2 + (1 - \gamma)0 \leq \frac{(dC\beta)^2}{\gamma}.$$

As in the proof of Theorem 4.1, for every iteration $t \geq 1$, let us denote $\mathbf{p}_{t+1} = \sum_{i=1}^{N} a_i \mathbf{v}_i$, $\bar{\mathbf{s}}_t = \sum_{i=1}^{N} a_i \mathbf{s}_i$, where $(a_1, ..., a_N)$, $\{(\mathbf{v}_1, \mathbf{s}_1), ..., (\mathbf{v}_{N_t}, \mathbf{s}_N)\}$ are the outputs of the call to Algorithm 2 on that iteration. Also define $\mathbf{p}_1 = \tilde{\mathbf{y}}_1$. Again, by the guarantee of Lemma 3.2, we have that

$$\mathbb{E}_{\{(b_t, i_t)\}_{t=1}^{T}} \left[ \sum_{t=1}^{T} \mathbf{p}_t \cdot \mathbf{f}_t \right] - \alpha \cdot \sum_{t=1}^{T} \mathbf{x}^* \cdot \mathbf{f}_t \leq \frac{\alpha^2 R^2}{\eta} + T \frac{\eta d^2 C^2 \beta^2}{2\gamma} + 3T\epsilon F.$$

Since $\mathbf{p}_t$ is dominated by $\bar{\mathbf{s}}_t = \mathbb{E}[\mathbf{s}_t]$ for all $t \in [T]$, we have that

$$\mathbb{E}_{\{(b_t, i_t, \mathbf{s}_t)\}_{t=1}^{T}} \left[ \sum_{t=1}^{T} \mathbf{s}_t \cdot \mathbf{f}_t \right] - \alpha \cdot \sum_{t=1}^{T} \mathbf{x}^* \cdot \mathbf{f}_t \leq \frac{\alpha^2 R^2}{\eta} + T \frac{\eta d^2 C^2 \beta^2}{2\gamma} + 3T\epsilon F.$$

Finally, since

$$\forall t \in [T]: \quad \mathbb{E}_{b_t}[\hat{\mathbf{s}}_t \cdot \mathbf{f}_t] = (1-\gamma)\mathbf{s}_t \cdot \mathbf{f}_t + \gamma \mathbf{q}_{i_t} \cdot \mathbf{f}_t \begin{cases} \leq \mathbf{s}_t \cdot \mathbf{f}_t + \gamma C & \text{if } \alpha \geq 1 \\ \geq \mathbf{s}_t \cdot \mathbf{f}_t - \gamma C & \text{if } \alpha < 1 \end{cases},$$

we have that

$$\mathbb{E}\left[\sum_{t=1}^{T} \hat{\mathbf{s}}_t \cdot \mathbf{f}_t\right] - \alpha \cdot \sum_{t=1}^{T} \mathbf{x}^* \cdot \mathbf{f}_t \leq \frac{\alpha^2 R^2}{\eta} + T\frac{\eta d^2 C^2 \beta^2}{2\gamma} + 3T\epsilon F + T\gamma C,$$

as required.

We now turn to upper bound the overall number of queries to the approximation oracle of $\mathcal{K}$. Note that we require to compute a new approximated projection only after rounds for which it holds that $b_t = \text{EXPLORE}$, since otherwise it holds that $\hat{\mathbf{f}}_t = \mathbf{0}$, and there is no update to the iterates maintained by Algorithm 3. For any $t \in [T]$ we define the indicator variable:

$$I_t \leftarrow \begin{cases} 1 & \text{if } b_t = \text{EXPLORE}; \\ 0 & \text{if } b_t = \text{EXPLOIT}. \end{cases}$$

Define $\hat{F} := \frac{dC\beta}{\gamma}$, and observe that for all $t \in [T]$ it holds that

$$\|\hat{\mathbf{f}}_t\| \leq \left\| \frac{d\mathbf{Q}^{-1}\mathbf{q}_{i_t}\ell_t}{\gamma} \right\| \leq \frac{d}{\gamma}|\mathbf{q}_{i_t} \cdot \mathbf{f}_t| \cdot \|\mathbf{Q}^{-1}\mathbf{q}_{i_t}\| \leq \frac{d}{\gamma}C\beta = \hat{F}.$$

Now, we continue to bound the number of calls to Algorithm 2, very similarly to the analysis in the proof of Theorem 4.1.

Let $k_t$ be the number of iterations it took Algorithm 2 to terminate when invoked on iteration $t$ of Algorithm 3 (w.l.o.g. this happens when Algorithm 5 sends the feedback $\hat{\mathbf{f}}_t$ to Algorithm 3), and note that $\mathbb{E}[K(\eta,\epsilon,\gamma)] = \frac{1}{T}\mathbb{E}\left[\sum_{t=1}^{T-1} k_t\right] \cdot O\left(d^2 \ln\left(\frac{(\alpha+1)R}{\epsilon}\right)\right)$. Note that for all $t \geq 1$, $\mathbf{y}_{t+1} = \tilde{\mathbf{y}}_t - I_t\eta\hat{\mathbf{f}}_t$. Thus, by Lemma 3.2, it follows that on any iteration $t$,

$$\begin{aligned} \text{dist}^2(\tilde{\mathbf{y}}_{t+1}, \text{CH}(\alpha\mathcal{K})) &\leq \text{dist}^2(\mathbf{y}_{t+1}, \text{CH}(\alpha\mathcal{K})) - (k_t-1)\epsilon^2 \\ &= \text{dist}^2(\tilde{\mathbf{y}}_t - I_t\eta\hat{\mathbf{f}}_t, \text{CH}(\alpha\mathcal{K})) - (k_t-1)\epsilon^2 \\ &\leq (\text{dist}(\tilde{\mathbf{y}}_t, \text{CH}(\alpha\mathcal{K})) + I_t\eta\hat{F})^2 - (k_t-1)\epsilon^2 \\ &= \text{dist}^2(\tilde{\mathbf{y}}_t, \text{CH}(\alpha\mathcal{K})) + 2I_t\eta\hat{F}\text{dist}(\tilde{\mathbf{y}}_t, \text{CH}(\alpha\mathcal{K})) + I_t\eta^2\hat{F}^2 - k_t\epsilon^2 + \epsilon^2. \end{aligned}$$

Rearranging, summing over all iterations $1...T-1$, and recalling that for all $t$, $\text{dist}(\tilde{\mathbf{y}}_{t-1}, \text{CH}(\alpha\mathcal{K})) \leq \sqrt{2}\alpha R$, we have that

$$\sum_{t=1}^{T-1} k_t \leq \frac{1}{\epsilon^2}\left(\text{dist}^2(\tilde{\mathbf{y}}_1, \text{CH}(\alpha\mathcal{K})) - \text{dist}^2(\tilde{\mathbf{y}}_T, \text{CH}(\alpha\mathcal{K})) + 2\sqrt{2}\sum_{t=1}^{T-1} I_t\eta\alpha\hat{F}R + \sum_{t=1}^{T-1} I_t\eta^2\hat{F}^2 + (T-1)\epsilon^2\right).$$

Taking expectation with respect to the random variables $I_1, ..., I_{T-1}$ we have that

$$\begin{aligned} \mathbb{E}\left[\sum_{t=1}^{T-1} k_t\right] &\leq (T-1)\left(1 + \frac{2\sqrt{2}\gamma\eta\alpha\hat{F}R + \gamma\eta^2\hat{F}^2}{\epsilon^2}\right) \\ &= (T-1)\left(1 + \frac{2\sqrt{2}\eta\alpha\beta dCR + (\eta dC\beta)^2/\gamma}{\epsilon^2}\right), \end{aligned}$$

as required.

$\square$