[Reviews · NeurIPS 2017]

Reviewer 1



This paper proposes online linear optimization approaches using approximation algorithms, where the authors consider the set of feasible actions is accessible. They explain how their approach captures natural online extensions of well-studied offline linear optimization problems. The paper is clearly written, easy to follow and addresses a very interesting problem. It seems to be theoretically sounds. However, the paper is very theoretical and neither uses synthetic or real data to validate the veracity of the formulated claims. And, I do not clearly see how these algorithms would perform with real world datasets. When we look at the paper from algorithms lenses we see that algorithm 1 is relatively classical (Online Gradient Descent Without Feasibility) easy to understand and accept theoretically, algorithm 2 ((infeasible) Projection onto...) seems to be convincing, however in practical situations they depend on a number of factors that are not necessarily captured/or cannot be captured by the alpha-regrets. To disprove this, it would be good to demonstrate that the proofs are congruent with some simulated data. I also believe that it would be good to demonstrate in a practical situation (even with simulated data) the performance of algorithm 3. Otherwise, the paper might not seem very compelling to the NIPS community.

Reviewer 2



The authors present a theoretical paper on online linear optimization with the application of online learning. The aim is to produce an algorithm which reduces computational complexity (aka number of required calls to an 'oracle', meaning a nearly-perfect optimizer - itself infeasible as NP but approximable, while providing a fast diminishing alpha-regret bounds, improving on computationally demanding procedures such as FPL. This paper assumes a constant (stationary) structure of adversary / expert action. The paper is entirely without numerical results (esp. comparisons to Abernethy et al.) and as opposed to oracle complexity (aka speed) the actual alpha regret bounds do not considerably improve on state-of-the-art: while theoretical bounds might be similar, oracle calls are not presumably simply 'wasted', so expected losses (average losses on a process) might still compare favorably for competing methods. In short, while this reviewer cannot refute the validity of the analysis or its relevance in the literature, feels that some experimental validation on well simulated data would be very useful in shedding further light on the exact loss/computational trade-off on offer. Otherwise the paper is well written, clear attention to detail, motivation, proofs etc and is quite commendable in that respect.

Reviewer 3



This paper addresses the problem of online linear optimization when the feasible set is such that doing exact projections into it is NP-hard but there is an approximate algorithm for maximizing linear functions in the feasible set with a constant approximation ratio. This paper is an extension to "Playing games with approximation algorithms", by Kakade et al with a similar algorithm with substantially better computational complexity. The main issue with the previous work (kakade et al) addressed in this paper is the linear (in the length of the online prediction history) number of calls to the approximation algorithm required in each online prediction round. Needing to essentially remember the entire history and have ever more expensive computation in each iteration is prohibitive and doesn't match most other work in online learning. This paper proposes a new infesible projection algorithm which given an input either returns its approximate projection into the feasible set or returns an infeasible point which is a convex combination of a known set of feasible points and is not too far from the input. Using this infeasible projection algorithm it's possible to make a stochastic online optimization algorithm which has better complexity (roughly because projecting a point which is near from an existing projection is easy). It is unclear what is applicability of this algorithm, however, as well as what would be the interest of it to the community. This seems to be of at most theoretical interest, and the paper should provide better justification of why this result is useful if it is to belong in nips.